# Relationship between Mentalizing and Working Conditions in Health Care

**DOI:** 10.3390/ijerph17072420

**Published:** 2020-04-02

**Authors:** Dagmar Steinmair, Felix Richter, Henriette Löffler-Stastka

**Affiliations:** 1Karl Landsteiner University of Health Sciences, University Hospital St. Pölten, Dunant-Platz 1, 3100 St. Pölten, Austria; Dagmar.Steinmair@gmx.net; 2Department of Psychoanalysis und Psychotherapy, and Teaching Center/Postgraduate Unit, Medical University of Vienna, Währinger Gürtel 18-20, 1090 Vienna, Austria; Felix-D-Richter@gmx.de

**Keywords:** psychotherapy training, reflective functioning, setting, sociodemographic, human-environmental perspective, working conditions

## Abstract

Mentalizing describes the human ability to comprehend one’s own and others’ mental states and is seen as one of the core competencies of psychotherapists. Current research has emphasized the importance of both early dyadic attachment as well as broader sociocultural environmental input on the development of mentalizing. This study investigates whether mentalizing skills, operationalized via reflective functioning (RF), might be influenced by training and working conditions. This study was a matched case-control comparison, cross-sectional study. RF was assessed in a total of 10 psychotherapy trainees working in private practice at the beginning (group A; *n* = 5) and end (group B; *n* = 5) of their psychotherapy training (training association: Gestalt Therapy, Institute of Integrative Gestalttherapy Vienna) and in a total of 40 health professionals (institution: General Hospital Vienna—Social Medical Center South, Vienna, Department of Psychiatry, acute psychiatric ward) at the beginning of (group C; *n* = 20) and without (group D; *n* = 20) mentalization based therapy training. The participants differed from each other regarding their training, but participants of the same institution were matched. RF scores were significantly higher in group A and B than in group C and D (A,C: *p* = 0.0065, Odds Ratio (OR): 0.0294; A,D: *p* = 0.0019, OR: 0.0132; B,C: *p* = 0.0065, OR: 0.0294, B,D: *p* = 0.0019, OR: 0.0132). RF scores were not significantly different among groups A and group B (A,B: *p* > 0.9999) or between groups C and D (C,D: *p* = 0.6050). The current study suggests that mentalizing skills might be rather slow to improve by training, but that they might be influenced by the context.

## 1. Introduction

Mentalizing describes the human ability to comprehend one’s own and others’ mental states and is seen as one of the core competencies of psychotherapists; the concept overlaps with the concepts of empathy [1,2]. Empathic abilities based on mentalization skills are a therapy factor with a known clinically relevant effect size on therapy outcome [3,4,5,6,7,8]. Mentalization based interventions have been shown to be important in the treatment of several mental disorders, including borderline personality disorder and other personality disorders [4,5,6,7,8]. Impairments in mentalization are associated with pathology of the self and with insecure attachment [9,10,11,12].

The theoretical knowledge about the need to enhance mentalizing abilities in psychotherapists opens the question whether empathic behaviour can be influenced, and if yes, how.

Previous research suggests that mentalizing is a highly interactive process and that environmental influences are crucial for its development [13,14,15]. Good mentalizing skills implicate an open and non-defensive mindset about the mental state of oneself and the others, leading to appropriate emotional responsiveness and emotionally regulated caring behaviour [14].

If the needs of the child are met consistently, a sense of predictability and a secure attachment pattern can evolve. The other can be acknowledged as a separate individual with autonomous affective and mental states. Reflective functioning is a concept introduced by Fonagy describing the ability to understand behaviour as implications of mental states [7]. Evidence seems to support the role of reflective functioning in the assessment of mentalization skills and the research field of attachment [6,9].

In addition to the known key role of the early dyadic relationship for the development of secure attachment and mentalizing skills [14,16,17], the influence of the whole social environment and context has been outlined lately by Fonagy et al. [15]. Attachment is supposed to be an innate, universal behavioural system [15,16], and to be a premise for the capacity of epistemic trust. Mentalization is required for deciphering the intentions of others and is necessary for filtering information provided by others with regard to their trustworthiness, which allows the individual to recalibrate thoughts and feelings when faced with adversity and to benefit from social learning [15,18], social contextual factors (like social media, relationships, culture), ongoing learning processes, and further influence the development of epistemic trust [15].

Relationships among attachment strategies, arousal, and mentalizing are still under investigation; however, evidence suggests that mentalizing skills may fluctuate considerably across contexts, even in persons with secure attachment and good mentalizing skills [15,19].

Real life settings pose various challenges to psychotherapists aiming to apply the concept of empathy; this emotional labor is requested independently of any aggravating circumstances such as patient characteristics, setting, work load, time pressure, unpredictability (i.e., lacking daily routine in emergency settings or during crisis management), job insecurity, and lacking career opportunities. “Extrinsic- and intrinsic-determinants of high-cost-low-gain situations at work have been shown to increase the risk of stress-related disorders, such as cardiovascular disease and depressive symptoms.” [20]. Further research has shown that demanding working conditions and negative stress perception influences the performance in health care workers [21,22].

Maladaptive coping strategies, such as over-commitment, may lead to work inability in the long run [20]. Budgetary cuts and the pressure on wards to reduce length of stay and making facilities available for psychiatric emergencies at all times requires thoroughgoing diagnostic assessments and therapeutic interventions in a cost and time efficient manner, without losing sight of the individual patient’s conditions and needs [23]. Despite ongoing efforts to improve the inpatient services, service user dissatisfaction and staff demoralization continue to be reported as problems [23,24].

For physicians, previous research has shown that high empathic abilities are a risk factor for emotional fatigue [25] and that empathy declines over time [26].

“Skills-focused training is not always directed toward fostering a genuine, strong, compassionate, caring relationship” [27].

The strength of the working alliance and the quality of the therapeutic relationship are known factors for better therapy outcomes in psychiatry [28,29]. There is evidence that the quality of the therapeutic relationship is afflicted by the setting and that acute psychiatric inpatient settings pose various challenges compared to regular scheduled therapeutic settings when trying to establish a good therapeutic relationship [24,29]. Bolsinger et al. [29] have emphasized that factors such as “involuntary admission”, “increased symptom severity”, “loss of autonomy”, “coercion” (perceived or actual), “role conflict in the therapist: Help vs. asses”, “team work, general setting” (i.e., loss of privacy, disrespect of patient’s preferences regarding characteristics of the therapist),“short duration, lacking continuity” (affecting the possible level of trust) can lead to a poor therapeutic relationship, especially in acute psychiatric inpatient settings [29]. Different patient characteristics might be a possible confounder for noticed differences between acute psychiatric inpatient vs. outpatient services [24]. Training in communication skills and raised situation awareness have been described as potential starting points for training programs, and the role of outpatient-directed services by involving individual social and environmental resources has already been outlined [29,30,31,32].

Required therapeutic techniques must not only be taught, but also their adequate implementation and responsive use in the particular therapy setting should be trained [33,34].

The aim of the current study was the assessment whether mentalizing skills, operationalized via reflective functioning (RF), can be influenced via setting/context or via training with strong focus on the development of such skills.

## 2. Materials and Methods

One part of the current study is part of the ongoing Society for Psychotherapy Research Interest Section on Therapist Training and Development (SPRISTAD) study, a multi-site longitudinal study of development in psychotherapy trainees [35,36,37]. The prerequisites for psychotherapy training candidates in Austria were considered and assessed within a survey at the beginning of their training. This background data was collected as part of the international study on training and professional development of psychotherapists (Society of Psychological Research Interest Section on Therapist Training and Development; SPRIS-TAD) with the Trainee Background Information Form (TBIF) [36]. The questions and subscales of the TBIF were taken from the questionnaire on the professional development of psychotherapists, which had been developed by the International Study on the Professional Development of Psychotherapists (ISDP) in 1989; an overview of the scales and their psychometric characteristics can be found in Orlinsky and Rønnestad [38]. Using 27 questions or subscales, sociodemographic data such as age, gender, nationality, education, and socio-economic status in the family of origin/childhood, as well as information about the educational institution in which the psychotherapy training is completed, were recorded [35].

The project received ethic approval from the ethics review board of the Medical University Vienna and the ethics committee of the City of Vienna.

### 2.1. Participants and Procedure

The aim of the current study was to assess whether mentalizing skills can be influenced via setting or via training with strong focus on the development of such skills. To evaluate this in a cohort of psychotherapy trainees and in a reference cohort, we invited two institutions to participate: A humanistic/experiential training institute (training association: Gestalt Therapy, IG Vienna; Institution 1) with a fitting definition of core competencies in their training program [39]; and a psychiatric inpatient clinic providing training in mentalization based therapy (MBT) for their employees on a voluntary basis (Institution 2).

Participants were recruited and assigned to four groups in the following way: A total of 58 psychotherapy trainees from Institution 1 and 61 employees of Institution 2 were invited to participate in the study via mail. A total of 5/27 psychotherapy trainees beginning their training at Institution 1 agreed to participate in the current study (group A) and were matched to 5/31 psychotherapy trainees who were at the end of their training (group B). A total of 40/61 employees of Institution 2 agreed to participate in the study (group C) and were matched to 20 employees who volunteered for further psychotherapy training and were at the beginning of a training in MBT at Institution 2 (group D).

The participants differed from each other regarding their training, but participants of the same institution were matched for age, gender, economic background, therapy experience, and family situation. Additionally, the participants of group C and D were matched for their previous psychotherapy training (e.g., kind of training, start/duration of psychotherapy training, “status” in terms of allowance to work with psychotherapy clients).

### 2.2. Procedure

The two samples from the humanistic/experiential training institute were compared with the two reference samples, and cross-sectional data of the four groups were observed and collected for the current study. All participants were asked to complete a questionnaire (TBIF) regarding their background data based on the questionnaire used in the SPRISTAD study, and were invited to an interview. The questionnaire for the beginner group (group A) was a 27 items online form assessing sociodemographic data (age, gender, nationality, education, socio-economic status during childhood). The questionnaire for the advanced group (group B) contained two more items (duration of the training, clients treated so far). The questionnaires were anonymized before sending them back to the study authors. The interviews were anonymized and sent to the rater without any further information about the participant. The interviews were rated using the reflective functioning scale (RFS).

The study participation was voluntary, participants all gave their written consent that their anonymized data could be used and published for scientific reasons. The participants did not get any financial compensation for participation.

### 2.3. Measures

Our primary outcome measure was the reflective functioning (RF) score.

The Reflecting Functioning Scale (RFS) is a measure for the capacity to mentalize thoughts, intentions, feelings and beliefs about oneself and others [1]. The Brief Reflective Function Interview (BRFI [40]) has been shown to be a reliable and valid screening measure of reflective functioning [41,42]. It was designed to detect severe impairments in mentalizing as a resource-friendly alternative to the Adult Attachment Interview [43] on which it is based [41,44]. The interview consists of 10 questions (demand and permit questions) exploring the respondent’s relationship with one of their parents.

RF of participants of all four groups was assessed using the BRFI. The interviews (*n* = 50) were rated and analyzed using the RFS, where a score of 1–3 indicated absent/low RF, a score of 4 indicated RF categories are identifiable in at least one passage of the interview, a score of 5 indicated moderate/ordinary RF, a score of 7 indicated marked/high RF, and a score of 9 indicated exceptional RF [1]. The overall reflective functioning categories were awareness of the nature of mental states, the explicit effort to separate mental states from underlying behaviour, recognizing developmental aspects of mental states, and mental states in relation to the interviewer.

All interviews were conducted in the mother tongue (German) of participants and rater and by the same rater for group A and B (Pleschberger I, certified RF coder 2017, German version) and by another rater for group C and D (Richter F, certified RF coder 2017, German version) in the same setting (Heidelberg/Hamburg; Trainer: Svenja Taubner). Both raters went through a certification procedure with several reliability tests resulting in accreditation and certification (for further information [45]). The interviews were rated by the same person who conducted the interviews after a seven month pause, after confirmed reliability of the rater/analyst.

The interviews were then compared to background data and were assessed with an adapted version of the Trainee Background Information Form (TBIF) designed by the SPRISTAD.

### 2.4. Data Analytic Strategy

All analyses were performed with Graphpad Prism8. We calculated frequencies of the achieved RF scores for each subgroup. For the comparison of RF scores of the four groups, we used Kruskal–Wallis test and calculated effect sizes. Because of the relatively small sample size and number of categorial variables, we used the Fisher’s exact test to analyze contingency tables for comparison of differences between the groups. For association between variables, we used Pearson’s correlation coefficient for normally distributed data and elsewise Spearman’s rank correlation coefficient.

## 3. Results

### 3.1. Demographical Characteristics of the Sample

For the descriptive statistic of the two psychotherapy trainee groups of the humanistic/experiential training program (group A: 5 beginner; group B: 5 training completed) and the two reference groups (group C: 20 employees beginning with MBT training; group D: 20 employees without MBT training) see Table 1.

As the four samples where matched, the distributions of age, gender, family situation, economic background in childhood and therapy experience were found to be not significantly different between the four samples (*p*-values: 0.17, 0.99, 0.18, 0.87, and 0.41 respectively; for Kruskal–Wallis test; see Table 1). Also having siblings and highest completed education were not significantly different across the four groups (*p*-values: 0.79 and 0.99 respectively). Having children differed significantly across the groups and percentage was highest in group C and D (65% and 85%, respectively). For all groups the demographic data were rather heterogeneous, reflecting the Austrian psychotherapy training requirement permitting trainees with a variety of professional backgrounds access to the training [35,36].

All groups show a higher percentage of female participants (62%) and the vast majority were married (76%), mean age across all groups was 43.28 years (standard deviation (SD) 9.783, min: 23, max: 61). The reported economical background during childhood was good in 54% of the participants, only one participant mentioned an insecure economical background.

### 3.2. Personal Therapy Experience and Training

All participants of group A and B participated either in personal therapy or in self-awareness programs. Personal therapy experience was highest in the practitioner group (group B; 80%) reflecting the high focus on personal self-awareness of the study program. In group D all participants except for one mentioned participation in self-awareness programs, and all of them started to attend MBT training, whereas most of the participants from group C did not complete a comparable self-awareness program (55%) and none of them were engaged in MBT training.

Achieved status was distributed significantly different across the four observed groups: group A: no status, group B: achieved status, group C: achieved status in 40%, group D: achieved status in 35%. Data on achieved status were missing in one participant of group C and in four participants of group D.

### 3.3. RF Measurements

For distribution of RF scores among the four groups see Table 2. Mentalizing skills operationalized via RF score did not differ significantly between the psychotherapy trainees at the beginning (group A) and the psychotherapy trainees at the end of the training (group B; Table 4). The RF scores of the group beginning with mentalization based training (group C) did not differ significantly from the group without this training (group D; Table 4). However, RF scores were found to be significantly different when comparing the four subgroups with each other (Table 2 and Table 4), with the highest mean RF scores in the two psychotherapy trainee groups (mean RF score group A: 7, group B: 6.2; Table 2). RF scores were significantly higher in group A and B than in group C and D (A,C: *p* = 0.0065, Odds Ratio (OR): 0.0294; A,D: *p* = 0.0019, OR: 0.0132; B,C: *p* = 0.0065, OR: 0.0294, B,D: *p* = 0.0019, OR: 0.0132).

Demographic data such as age, gender, family situation, having children and the economic background did not show an influence on RF scores (Table 3) in our sample, as no significant correlation was found. Admittedly, only one participant with an insecure economic background in childhood was included in our investigation and they achieved an above average RF score (RF score: 6; average RF score: 4,49; SD 1.86). The vast majority of participants reported a good (54%), adequate (30%), or very good (14%) economic background in childhood.

Working full time correlated significantly and positively with higher RF scores (level of employment in Table 3) for probands working in the inpatient clinic (for the probands in the outpatient clinic this data was missing). Working in a private praxis was associated with a significantly higher RF (setting in Table 3).

In our investigation the factors “being in psychotherapy training” (i.e., engaged in the psychotherapy propedeutics, attending one’s own training therapy) or having achieved the psychotherapy status to work with clients were significantly associated with higher RF scores, suggesting higher mentalizing skills in participants with interest in psychotherapy training/psychotherapy background (Table 3).

Fisher’s exact test showed that RF was significantly higher in group A and B than in group C and D (A vs. C: *p* = 0.0065, OR: 0.0294; A vs. D: *p* = 0.0019, OR: 0.0132; B vs. C: *p* = 0.0065, OR: 0.0294, B vs. D: *p* = 0.0019, OR: 0.0132; see Table 4). RF scores were not significantly different among group A and group B (A vs. B: *p* > 0.9999) or between group C and D (C vs. D: *p* = 0.6050).

## 4. Discussion

In the present study we investigated mentalization capacity and mentalizing skills measured via BRFI in health professionals. Based on the body of literature suggesting that mentalization skills might be influenced and formed by social interactions and by the whole environment [13,14,15], we expected that setting and training might have measurable influences on RF scores even in mentally healthy health care workers.

Our results suggest, that RF scores are relatively stable over time and might not be influenced significantly by psychotherapy training in this sample. This might be consistent with the evidence of patterns of adult attachment that are also supposed to be relatively stable [46], and with previous research showing that reflective functioning mediates the relationship between attachment and personality [47]. Human attachment relationships are regarded as mediators in establishing the human ability for self-consciousness. In light of this assumption, a new conception of personality has been established based on temperament, mentalization, and attachment [48]. There is evidence that mentalizing skills are intergenerationally transmitted and they are seen as an evolutionary adaptation to the need of human beings’ shared intentionality and cooperation [15,49].

However, previous research provides evidence for fluctuations of mentalizing skills in borderline disease patients and also in non-clinical settings of healthy probands [15].

The mean RF score in our study sample was 4.49 (SD = 1.86), corresponding to a slightly below normal RF score (ordinary RF score = 5). This finding is consistent with previous studies supporting evidence for the decline of medical students’ empathy during training [50,51,52,53,54]. Physicians with a higher degree of empathy are even more prone to emotional fatigue [25].

The highest mean RF scores were found in the two psychotherapy groups working in private practice, suggesting significantly higher mentalizing skills (see results, Table 2, Table 3 and Table 4). In previous studies, attachment styles of psychotherapy trainees were different when compared with participants of two non-clinic reference samples, where attachment avoidance was reported to be lower in psychotherapy trainees, attachment anxiety was shown to be similar in all samples, and interpersonal motives (i.e., feeling comfortable in close relationships, achieving cooperation) were more pronounced in psychotherapy trainees [55].

In our study working full-time was associated with higher RF scores than working part-time. This might be the effect of higher patient numbers, and hence gaining more practical experience, suggesting that mentalizing skills might be trainable. As the causes of downshifting were not analyzed in our study, there might be unknown confounders.

Working in private practice was associated with higher RF scores when compared with working in an inpatient clinic (*p* < 0.0001; Table 3 and Table 4). Either persons with higher mentalization skills as a prerequisite for self-consciousness tend to prefer working in private practice, or the care setting and organizational variables (social context, working conditions, working hours, patient selection) might influence the missing further development of RF skills in the inpatient sample. Our results seem to be not in line with previous research showing a lack of influence of the setting on clinical empathy of physical therapists [56]. Nevertheless, it has to be mentioned, that the groups (especially Group A and B) consist of a small number, therefore data analysis does not allow strong conclusions. Our discussion has to be considered more as assumptions that can help guide hypotheses for further research.

Although mentalizing has been shown to be relatively stable, contextual factors are known to have an influence, such that increasing arousal and stress may lead to fluctuations [15]. Previous research has indicated that external factors impact on mentalizing abilities, where stress and negative stress perception have a negative impact on empathy, whereas social support and career satisfaction were found to have a beneficial effect on empathy measures [50,57]. Empathy is not correlated with personality traits [51]. The consequences of stress exposure on dehumanization behaviour (i.e., aggressive behaviour offending peoples’ dignity) in medical professionals has been shown to be independent of empathy scores, neuroticism, and psychoticism [58].

Severity of stress resulting from exposure to a traumatized individual (i.e., compassion fatigue) has been shown to be aggravated by the severity of the pathologies the caregiver has to deal with [59,60]. In our study the inpatient practitioners were generally exposed to patients with more severe and acute psychopathology, such that mean global assessment of functioning score of patients at Institution 2 was 30, whereas it was 70 for patients at Institution 1.

Cognitive functioning in general does not predict perception of stressful working conditions [20]. Mentalization plays a role in dealing with subjective and interpersonal distress and perception of distress might be influenced by the attachment pattern [4,61].

Mentalizing involves cognition (perspective-taking, belief–desire reasoning), affect (felt reality), and the integration of both [15]. In the face of arousal and challenges, Luyten et al. assume there is a shift from controlled (conscious, reflective, verbal) to automatic (reflexive) mentalizing, where the threshold for this switch and for recovery to controlled mentalizing depends on the development of individual mentalizing skills [15].

Perceived threats activate the attachment system within the framework of stress response and fear appraisal [62]. Early experiences in this interconnected interpersonal regulation systems impact on the development of mentalization skills and on the vulnerability to further stressful events [61]. RF seems to improve with age, such that when confronting adolescent vs. adult mother–infant dyads, adolescent vs. adult mothers had a lower RF score [63].

The early dyadic attachment has been shown to be critical in the establishment of mentalizing and epistemic trust, but current views emphasize the role of interactions with relevant others and with the whole sociocultural environment [15]. Resilience to distress and challenging circumstances is engrained in secure attachment and solid mentalizing skills. Mentalization is a highly interactive process and “it develops in the context of interactions with others, and as a result it is assumed to be continually influenced by the mentalizing capacity of those others” ([15], p8). In this context, attachment measurements of our sample could be interesting, and are currently planned for a further study in a larger sample. Nevertheless, it has been shown already that attachment styles in psychotherapists are distributed quite similar or equally to the normal population, and that effects of training therapy and training of self-awareness have to be taken into account for further investigations [64].

The neurobiology of stress response is still under investigation. Adaptive reactions to acute stress include shifts in the large-scale brain networks (salience (SN), central executive (CEN), and default mode networks (DMN)) under tasks and without ongoing task demands [65]. For children, anxious and uncontained parenting was associated with higher cortisol levels and with reduced DMN connectivity in preadolescence [66]. Attachment-related stress also has specific effects on mentalizing in adulthood [62,67]. Previous research has investigated the effects of stress on the neural correlates of mentalizing in adults, whereby the activation of an attachment-specific stress system modulates the functioning of brain regions involved in mentalizing (left posterior superior temporal sulcus, left inferior frontal gyrus, left temporoparietal junction) [67].

Neuroimaging research has shown that compared to healthy controls, posttraumatic stress disorder patients have hyperactivation of default mode network (DMN) and mirror neuron system (MNS) to emotional stimuli [68]. Impairment of DMN connectivity has been shown to be associated with impairment of the self’s experience in psychopathology [69].

Emotional exhaustion and health complaints are consequences of stress, and affect workers focused on helping others as they often lead to job tenure and contract breaches [60].

When basic expectations during childhood are met and a secure attachment pattern evolve, epistemic trust and self-worth give the individuals confidence in exploring their environment and in acquiring social knowledge [1]. This might be one precondition for arranging working conditions to one’s own needs, perhaps by taking the freedom to create a working environment at one’s own option and/or by making self-care a first priority.

Engagement of mental health practitioners in self-care has been shown to reduce stress related psychopathology and to improve their work, such that the care setting might influence possibilities and obstacles in establishing continuous and adequate self-care practice [70]. Social and collegial support, a supporting infrastructure, and positive affirmation have been shown to positively influence the capacity to cope and in prevention of compassion fatigue in nurses and in medical students [57,71,72].

Achieving knowledge and skills in the diagnosis and treatment of mental disorders does neither protect from mental disorders nor guarantee mental health. Quite the contrary, investigations on serious emotional problems and vulnerability for suicide among psychotherapists reported an even higher suicide risk in this population, possibly due to stress and strains in this profession or due to a self-selection or unknown characteristics of psychotherapy trainees [73,74]. It is in the nature of things that psychotherapists, as with others engaged in helping professions, deal with high demands on their resources, but furthermore they are expected to be “paragons” when handling with stressors such as emotional and behavioral crises arising within their professional and private life. This makes them a “prime target for disillusionment, distress, and burn out” [75].

However, research indicates different empathy styles amongst therapists, and the fashion in which empathy is practiced could impact on coping abilities [76]. Previous research indicates a four profile solution for empathy amongst therapists: (1) insecure/self-absorbed (concerned with own reactions to patients; 23%), (2) empathic immersion (intuiting and identifying; 26%), (3) average (38%), and (4) rational empathic (intellectual understanding and perspective taking without identifying with the patients experience; highly regulated; 13%) [76]. Previous research on empathy suggests that trainees with higher empathy at the beginning of training improved their level of empathic communication more successfully than trainees with lower baseline skills [77].

This has implications for the design of future psychotherapy training curricula, whereby self-care and personal therapy should be given priority by including a baseline assessment of personality traits with a strong focus on mentalizing and empathy profiles and attachment styles.

Obstacles for seeking help should be minimized in psychotherapy trainees working environments, and supervision and self-assessment should be emphasized. The Hippocratic sentence “primum non nocere, secundum cavere, tertium sanare” should be first considered when dealing with oneself just as much as when dealing with clients.

### Limitations

We only analyzed cross-sectional data. Further investigation should analyze time-series and follow-up data of trainees. It would be interesting to relate assumed mentalization skills to observations of interpersonal behavior in therapy sequences and to patient therapy outcomes.

Our findings may not be generalizable to other cultures and health care systems as all of our participants and raters were living in Austria with German as the mother tongue.

Another limitation of the study is the disbalance between the sample size of therapists working in the acute psychiatric ward vs. in private practice, and the quite small number of participants for the latter. Critically, the selection of therapy trainees must also be mentioned, as the trainees agreeing to complete the study might be different from the trainees avoiding such investigations.

The reflective functioning was assessed with the BRFI. A comparison with other measures of reflective functioning [19], measuring even mild impairment, would be desirable in future trainees and practitioner groups as severe pathologies in these samples are supposed to be relatively seldom, and subtle improvements of mentalization skills might be difficult to detect.

Research on RF is still very limited, further research should focus on its applicability in the field of psychotherapy research [6].

## 5. Conclusions

To what extent can psychotherapeutic skills be developed? Empathy and psychotherapy outcome are significantly associated, and empathy is shown to be the specific therapy factor with the largest effect size. Hence psychotherapy training is supposed to provide training in empathy and in the ability to constructively utilize empathic skills in clinical work.

Based on our findings we cannot provide full evidence for the successful development of empathic skills during psychotherapy training as mentalizing skills might be rather stable over time. Nevertheless, psychosocial working conditions and challenging settings might influence mentalizing skills in a more significant fashion.

Self-care and personal therapy should be given priority during psychotherapists’ training, as working as a psychotherapist requires strong crisis response abilities and prevention of compassion fatigue.

## Figures and Tables

**Table 1 ijerph-17-02420-t001:** Demographical characteristics of the sample.

Sociodemografic Variables			Group A	Group B	Group C	Group D	All	H	*p*-Value
		*n*	5	5	20	20	50		
Gender	w	*n* (%)	3 (60)	3 (60)	13 (65)	12 (60)	31 (62)	0.1248	0.9887
	m	*n* (%)	2 (40)	2 (40)	7 (35)	8 (40)	19 (38)		
Age		Mean	37	35,4	45,35	44,75	44,8	5.061	0.1673
		Max	46	46	61	58	61		
		Min	27	23	28	31	23		
		Median	41	37	45	43,5	44		
Family situation	married	*n* (%)	4 (80)	4 (80)	17 (85)	13 (65)	38 (76)	4.859	0.1824
	divorced	*n* (%)	1 (20)	1 (20)	3 (15)	2 (10)	7 (14)		
	no	*n* (%)	0	0	0	5 (25)	5 (10)		
Children	yes	*n* (%)	3 (60)	2 (40)	13 (65)	17 (85)	35 (70)	8.685	**0.0338**
	no	*n* (%)	2 (40)	3 (60)	7 (35)	3 (15)	15 (30)		
Siblings	yes	*n* (%)	4 (80)	4 (80)	14 (70)	16 (80)	38 (76)	1.034	0.793
	no	*n* (%)	1 (20)	1 (20)	6 (30)	4 (20)	12 (24)		
	>1	*n* (%)	2 (40)	3 (60)	11 (55)	12 (60)	28 (56)		
		mean	1.4	2.2	2.10	2.7			
Economic background in childhood	insecure	*n* (%)	0	1 (20)	0	0	1 (2)	0.698	0.8737
	adequate	*n* (%)	2 (40)	1 (20)	6 (30)	6 (30)	15 (30)		
	good	*n* (%)	3 (60)	2 (40)	11 (55)	11 (55)	27 (54)		
	very good	*n* (%)	0	1 (20)	3 (15)	3 (15)	7 (14)		
Status (working with patients under supervision) °	yes	*n* (%)	5 (100)	0	8 (40)	7 (35)	20 (40)	10.07	**0.018**
	no	*n* (%)	0	5 (100)	11 (55)	9 (18)	25 (50)		
	missing data	*n* (%)	0	0	1 (5)	4 (8)	5 (10)		
Propedeutics *	yes	*n* (%)	5 (100)	5 (100)	8 (40)	9 (18)	27 (54)	9.997	**0.0186**
	no	*n* (%)	0	0	12 (60)	11 (55)	23 (46)		
Advanced training in psychotherapy *	yes	*n* (%)	5	5	8 (40)	9 (18)	27 (54)	9.997	**0.0186**
	no	*n* (%)	0	0	12 (60)	11 (55)	23 (46)		
MBT *	yes	*n* (%)	0	0	0	20 (100)	20 (40)	49	**<0.0001**
	no	*n* (%)	5 (100)	5 (100)	20 (100)	0	30 (60)		
Therapy experience	yes	*n* (%)	3 (60)	4 (80)	12 (60)	12 (60)	31 (62)	2.866	0.4127
	no	*n* (%)	2 (40)	1 (20)	8 (40)	8 (40)	19 (38)		
Highest education	matura	*n* (%)	2 (40)	1 (20)	1 (5)	4 (8)	8 (16)	0.0975	0.9921
	college	*n* (%)	0	2 (40)	3 (15)	2 (10)	7 (14)		
	DGKP	*n* (%)	0	0	6 (30)	4 (8)	10 (20)		
	uni	*n* (%)	3 (60)	2 (40)	9 (18)	10 (50)	24 (58)		

H: Kruskal–Wallis statistic; Group A,B,C,D: see methods; ° status: qualification to work with clients; *: participants entering training propedeutics (i.e., pre-university pathway course), advanced studies/training in psychotherapy (German: Fachspezifikum) and MBT (i.e., mentalization based therapy training) respectively; DGKP: college of nursery.

**Table 2 ijerph-17-02420-t002:** Reflective functioning.

Reflective Functioning		Group A	Group B	Group C	Group D	Totals	H	*p*-Value
*n*		5	5	20	20	50	20.83	**0.0001**
RF score	<5	0	0	15	12	27		
	≥5	5	5	5	8	13		
	Mean	7	6.2	3.5	4.4	4.5		
	Max	8	7	7	7.5			
	Min	5	5	0	2.5			
	Median	7	6	3.5	4.45			

RF: reflective functioning score; H: Kruskal–Wallis statistic; Group A,B,C,D: see methods; group A,B psychotherapist trainees (Institution 1); group C,D reference group (Institution 2).

**Table 3 ijerph-17-02420-t003:** Reflective functioning: confronting RF scores with participants characteristics.

Participants Characteristic	*p*-Value
Age	0.8898
Gender	0.6587
Nationality	0.0554
Educational background	0.1486
Economic background °	0.0741
Setting	<0.0001
Level of employment	0.0366
Propedeutics *	<0.0001
Fachspezifikum *	<0.0001
Status	0.0359
Therapy experience	0.0006

* participants entering training (propedeutics, Fachspezifikum); status: qualification to work with clients; ° economic background in childhood.

**Table 4 ijerph-17-02420-t004:** Reflective functioning: Comparing RF scores among the four groups (A–B).

RF Score	*p*-Value	OR
RF-A vs. RF-B	>0.9999	
RF-A vs. RF-C	0.0065	0.0294
RF-A vs. RF-D	0.0019	0.0132
RF-B vs. RF-C	0.0065	0.0294
RF-B vs. RF-D	0.0019	0.0132
RF-C vs. RF-D	0.6050	

RF-A–D: mean RF score Group A,B,C,D: see methods; group A,B psychotherapist trainees (Institution 1); group C,D reference group (Institution 2); OR: odds ratio.

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
