# Peer review of "Relationship between Mentalizing and Working Conditions in Health Care"

_ijerph, 2020, doi:10.3390/ijerph17072420_

Round 1

Reviewer 1 Report

Interesting topic: can mentalizing skills be improved. Sample is a bit on the small side.

Good English.

The paper needs quite a bit of work.

ABSTRACT

Well-written abstract, gives one a clear idea of what the paper is about. However, a crucial piece of information is missing: the effect size. How much are the people increasing and is this a relevant increase. The manual of the American Psychological Association states clearly that effect sizes should always be reported and that conclusions of studies should be mainly be built upon them.

INTRODUCTION

The beginning of the first paragraph is quite unclear. Several questions are being asked and then all of a sudden, the authors further discuss one of the questions. This is a quite unusual way to start a paper.

Introduction needs a lot of work. It is too short. It is difficult to read and follow. There is very little structure. The research question does not logically follow from what is written before.

A lot of topics are touched upon, without forming a good narrative.

MATERIALS AND METHODS

Supply some more details on the SPRISTAD study. Will not be familiar to all readers.

  1. 61 Approval from which university? Please be specific.
  2. 63-64 This kind of is the Research Question and it should be worked out in Introduction, not in Method.
  3. 68 mentalization-based therapy; don’t forget your hyphens if you want a clear text. Various hyphens missing in the rest of the manuscript.

Good matching procedure.

  1. 99-100 This kind of information belongs in Introduction, not in Method.

L 118 7-months pause; don’t forget your hyphens

  1. 121 Supply more details on the TBIF.

RESULTS

Table 1. Some of the information will not be comprehensible for non-Germans, so supply descriptions or translations.

There is a fundamental flaw in this section: A total reliance on significance testing and ignoring of effect sizes. Fundamentally rewrite this section.

  1. 159 100% reliance on significance testing. Cohen (1988) describes how to interpret sizes of correlations.

Generally: competently analyzed

DISCUSSION

Please explain how a paper which such a modest sample size contributes sufficiently to the literature.

Fundamental flaw: Conclusions based solely on significance levels. This is a big problem as the sample size is clearly modest. Fundamentally rewrite.

Formulate specific hypotheses on the effect sizes to be expected based on the literature, in Introduction.

The length of Discussion and the length of Introduction are out of balance, and strongly so. Discussion is much too long in comparison to the too short Introduction. Bring them into balance.

ENCOURAGEMENT

Good luck with the revision.

Author Response

Dear reviewer 1, many thanks for your helpful suggestions, our answers see in the attached file, kind regards, Henriette Löffler-Stastka

Reviewer 2 Report

The study in itself is interesting but the groups have a low number, data analyzes are few and no strong conclusions can be drawn. The organization of the different parts of the article must be reviewed.

The introduction is very short and must be improved.

In the measures the authors should indicate the agreement of the judges.

It would also be interesting to evaluate the attachment of therapists.

Why did the authors decide to use a categorial score of RF and not the score on 0-9 points? They could use an Anova

The groups A and B had a lower size than groups c and d

With these number of partecipans the discussion should be based more on assumptions than on clear conclusion.

In the discussion the part of theoretical study should be removed and should be inserted in the introduction and the authors should be extened the concept of RF and the low RF in a at risk sample (see for example Riva Crungola et al., 2018.Reflective functioning, maternal attachment, mind-mindedness, and emotional availability in adolescent and adult mothers at infant 3 months).

In the limit, the authors should be write that the sample size for each group is very low

Author Response

Dear reviewer 2, many thanks for your suggestions, the answers are provided in the attached file, many thanks!

Round 2

Reviewer 1 Report

The authors seriously and conscientiously processed my feedback. The paper now looks more solid and most likely will be read and cited more often than the previous version.

Reviewer 2 Report

The authors have improved the manuscript following the suggestions